# Underwater 2D Image Acquisition Using Sequential Striping Illumination

**Benxing Gong * and Guoyu Wang**

College of Information and Engineering, Ocean University of China, Qingdao 266100, China;
gywang@ouc.edu.cn
* Correspondence: gongbenxing@stu.ouc.edu.cn

**Abstract:** Structured lighting techniques have increasingly been employed in underwater imaging, where scattering effects cannot be ignored. This paper presents an approach to underwater image recovery using structured light as a scanning mode. The method tackles both the forward scattering and back scattering problems. By integrating each of the sequentially striping illuminated frame images, we generate a synthesized image that can be modeled on the convolution of the surface albedo and the illumination function. Thus, image acquisition is issued as a problem of image recovery by deconvolution. The convolutional model has the advantage of integrating the forward scattering light into a recovered image so as to eliminate image blur. Notably, the removal of the back scattered light from each frame image can be easily realized by a virtual aperture to limit the field of view; the same principle as of the synchronous scanning systems in underwater imaging. Herein, the implementation of the proposed approach is described, and the results of the underwater experiments are presented.

**Keywords:** structured light; scanning imaging; back scattering removal; synthesized image; image recovery

## 1. Introduction

As one of the essential 3D measurement technologies, structured lighting is increasingly being applied in underwater imaging; those frequently required include, seabed mapping, pipeline or dock detection, target localization, and ROV (remotely operated vehicles)/AUV (autonomous underwater vehicles) navigation. A variety of structured lighting techniques have been developed for these applications, such as laser line scan systems [1], 3D laser line scan mapping systems [2], laser line sensors [3], and seabed-relative navigation by structured lighting techniques [4]. In general, structured lighting techniques modulate the appearance of a surface by projecting a particular pattern of light onto it [5]; subsequent correspondence of the image pattern and surface points, provided with calibration of the image sensor and projector, enables accurate computation of the range of surface points. Structured lighting techniques can be classified as multiple-shot or single-shot categories. The multiple-shot technique works with a light stripe scanning mode where the striping illumination is sequentially swept across an object (Figure 1), by which the 3D computation is completed frame by frame. The single-shot technique projects a complex coded illumination pattern onto the objects, and the 3D computation is completed from a single image by decoding the reflected surface light pattern. However, optical imaging in scattering media suffers from scattering problems [6]. The back scattered light, which is scattered back from the medium before projecting onto the object surface, veils the scene and degrades the image contrast. The forward scattered light, which comes from the illuminated object surface but deviates during the transmission to the camera, blurs the details of the scene and degrades the sharpness of the image. Thus, an optical image taken underwater usually loses meaningful image features of scenes. In practice, most single-shot techniques are inapplicable in turbid water medium,

because the coded lighting pattern is hard to detect, with very low contrast caused by the back scattered light. Thus, in many applications, structured lighting techniques using sequential striping illumination is a reliable implementation of underwater 3D measurement, in terms of time and related cost [7].

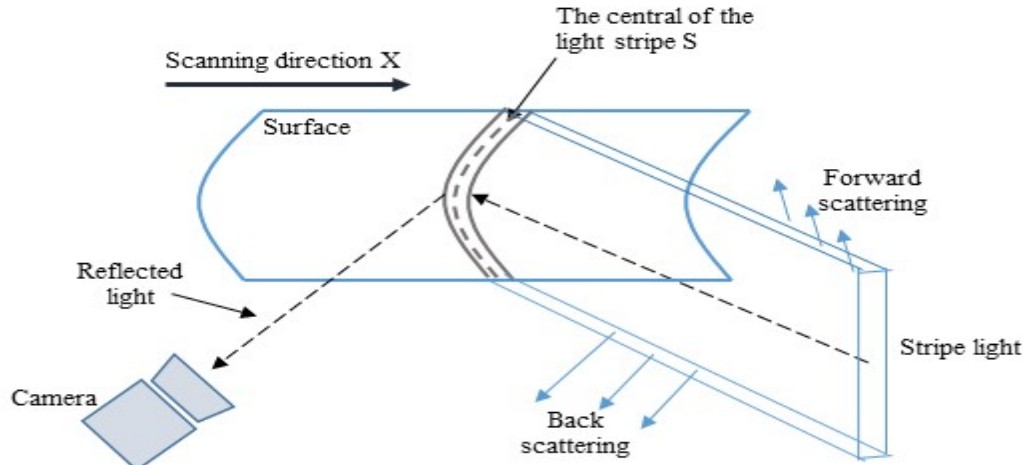

**Figure 1.** Typical structured lighting technique.

Acquiring a 2D image while 3D measurement is implemented using the structured lighting illumination underwater, is highly desirable. Although additional flood lighting devices can induct extra deployment of hardware, the desire is mainly due to the fact that the sequential striping illumination modality is able to remove the back scattered light. For the removal of the back scattered light, a well-known approach is the physics-based gating technique, a typical example of which is the synchronous laser line scanning system (SLL) [8,9]. The system works with the laser line (or stripe) illumination and the aperture-coupled opto-electric sensor, both of which scan synchronously. As the overlapped volume of the illuminated field of the laser and the view field of the aperture is very limited, the amount of the back scattered light received by the sensor is also very limited; most of the back scattered light generated outside the overlapped volume is spatially separated from the reflected light of the object. The image pixels are specified by stitching the sequentially scanned outputs of the opto-electric sensor. The SLL system has been deeply investigated, but only in terms of the cost of hardware complexes. Alternatively, a compact device consisting of a CCD camera and a structured light source can facilitate underwater image acquisition, albeit incorporated with 3D computation. In [10], a comprehensive analysis of the light stripe range scanning method is given. Based on the assumption of single scattering, an analytic image formation model is derived to describe the interactions of light with the medium and the scene. Using the image formation model for light striping, a simple algorithm is developed to obtain the 3D reconstruction of the scene in the presence of strong scattering. For calibration, the 3D world coordinates of a few points on planes are measured a priori, and the parameters describing the optical properties of the medium are estimated by fitting observations to the model. Once the attenuation coefficient is evaluated and the 3D surface is reconstructed, the clear-air appearance of the scene can be computed for each object intersection strip and all the intersection strips geometrically stitch together to generate the 2D image. The algorithm relies on the evaluation of the attenuation coefficient and the preset calibration, which required extra in situ measurements underwater.

The image acquisition using structured light has been given less attention until now. It is clear that the existing methods mentioned above work for the removal of back scattered light but still neglect the forward scattering effect. In their methods, image acquisition commonly utilizes the stitching approach, i.e., simply stacking the sequential frames to output the 2D image. Although stitching operation is straightforward, it fails to tackle the forward scattering effect in image acquisition, as stated in Section 2.

In this paper, we present a new approach to image acquisition using a structured lighting source and a CCD camera. Significantly, both forward scattering and back scattering are dealt within the image acquisition. Using the striping illumination in a scanning mode, sequential frame images are collected by the CCD camera. Instead of deploying an opto-electric coupled aperture sensor in the SLL system, we apply a virtual aperture to limit the field of view in each frame image, so as to separate the back scattered light from the reflected light of the object. The virtual aperture is specified by a masking operation, having the same mechanism as the SSL system [9] but computationally implemented rather than physically implemented. Different to the stitching approach commonly used in previous work for image formation, a method of image recovery is proposed. Sequential frame images are modeled on the convolution integration of the illumination function and the albedo of the object surface, with which each frame image is processed by integration operation. Thus, the albedo of the object can be recovered by deconvolution. The advantage of the proposed algorithm is that the forward scattering effect is fully eliminated through the integration operation. Theoretically, the integration operation enables us to collect the forward scattered light generated during the transmission from the object to the camera, enhancing the total imaging energy of the object.

## 2. Approach of Image Recovery from Sequential Striping Illuminated Images

### 2.1. Image Formation Model

A typical structured lighting technique is illustrated in Figure 1. The projector projects a light plane and sweeps across the object surface. By detecting the image position of the intersection curve of the light plane and the surface, the 3D coordinates of each point on this curve can be computed once the correspondence between the CCD coordinate and the projector coordinate is set up.

It should be noted that, in real implementations, stripe lighting refers to a stripe-like illumination whose transverse intensity distributions have a very narrow support. Thus, the light plane is just an approximated model of structured lighting. Instead of the geometrical model of a light plane, we consider the transverse distribution of a light stripe projected onto an object surface, observed from the CCD image coordinate $I(x)$, where $x$ is situated along the scanning direction (Figure 1). Thus, we can represent the image formation with respect to a one-dimensional description associated with a single image row along the $x$ coordinate. Suppose the albedo of the object surface is $O(x)$. At a moment during the scanning duration, the center of the light stripe is positioned at $s$. Thus, the one-dimensional image with respect to a change of $s$ can be expressed as:

$$f(x,s) = I(x-s)O(x) \qquad (1)$$

In underwater imaging, however, the forward scattering occurs during light transmission from the object to the camera, whose imaging effect can be expressed by the point spread function (PSF) $h(x)$, whereas back scattering induces added background component $b(x)$. Thus, the general image formation is:

$$f(x,s) = h(x) * [I(x-\tau)O(x)] + b(x,s) \qquad (2)$$

where $*$ is the convolution operation. We now take the integration operation on both sides of Equation (2) with respect to the variable $x$. We have:

$$g(s) = \int_{\Omega} f(x,s)dx = \int_{\Omega} h(x) * [I(x-s)O(x)]dx + \int_{\Omega} b(x,s)dx \qquad (3)$$

where $\Omega$ is the row image domain. Noting the equivalent expressions:

$$\int_{\Omega} h(x)dx = 1 \text{ and } \int_{\Omega} h(x) * f(x)dx = \int_{\Omega} f(x)dx \qquad (4)$$

We can formulate Equation (3) as:

$$g(s) = \int_{\Omega} I(x-s)O(x)dx + \int_{\Omega} b(x,s)dx = I(s) * O(\tau) + B(s) \tag{5}$$

In the above derivations, we have implied that $I(x)$ is of symmetrical distribution. The form of Equation (5) suggests that the function $O(x)$ can be obtained by the deconvolution operation once we generate the image g(s) by integrating each of the sequentially striping illuminated frame images. Consequently, the albedo image is obtained by the recovered $O(x)$ row by row.

Some limiting factors have to be addressed to apply the model of Equation (5). It is noted that the illumination function $I(x)$ is actually not the same as that originated from the projector. The function $I(x)$ is the projected light distribution observed from the view of the camera, whose distribution is relevant to the surface shape and is affected by forward scattering during projection. It implies that the estimation of $O(x)$ should be implemented through blind deconvolution by Equation (5), for which the exact distribution of $I(x)$ is unknown from the observation of the camera. Obviously, however, the first illumination distribution from the projector provides a good initial estimate of $I(x)$. In addition, attenuation during light transmission has to be considered in image formation. We omit this in Equations (1)–(5) because the attenuation factor, exp(-cl), where $c$ denotes attenuation coefficient and $l$ the transmission length, is a multiplicative scalar that can be merged into the function of $O(x)$.

It should be noted that, instead of integration with respect to $x$ in Equation (2), the integration with respect to the variable $s$ gives rise to the stitching result, which is expressed as:

$$g'(x) = K \cdot h(x) * O(x) + B'(x) \tag{6}$$

where $K = \int_{\Omega} I(x)dx$. Thus, strictly speaking, the stitching result is a blurred version of the object $O(x)$ due to the forward scattering effect of $h(x)$. Instead, the proposed approach modeled on Equation (5) collects the forward scattered light through integration of $h(x)$ and represents image acquisition as a deconvolution operation; the illumination pattern $I(x)$, whose prototype is explicitly designed at the projector, is estimated simultaneously.

### 2.2. System Implementation

The system implementation is illustrated in Figure 2. The DLP (Digital Light Processing) projector is used as the illumination source. The CCD video camera (GO-5000-PGE produced by JAI company, Denmark.) is controlled to record the frame images synchronously while the illumination scans across the object. The projected illumination pattern scans along the direction aligned with the x coordinate of the CCD image.

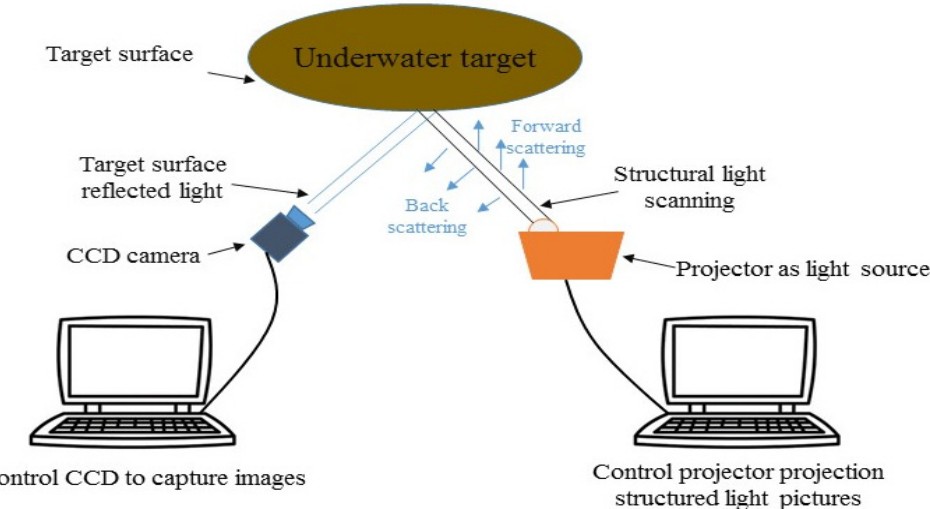

**Figure 2.** System implementation.

The illumination pattern *I(x)*, specified in (5), is designed on the projector and the scanning is realized by sequentially translating the center of the illumination pattern on the projector imaging plane to a fixed displacement (1 pixel in our experiment), which is controlled through a computer program.

Concerning the requirements of 3D measurement, the transverse distribution of the lighting stripe, generated from the projector, is designed to be a Gaussian distribution, so the shape of projected *I(x)* could facilitate detection of the center of the structured light in 3D computations. The width of the stripe, the parameter describing the normalized Gaussian distribution s, was set to 9 pixels.

A schematic description of the sequential illumination process and an example of the sequential frame images obtained from a real object are illustrated in Figure 3.

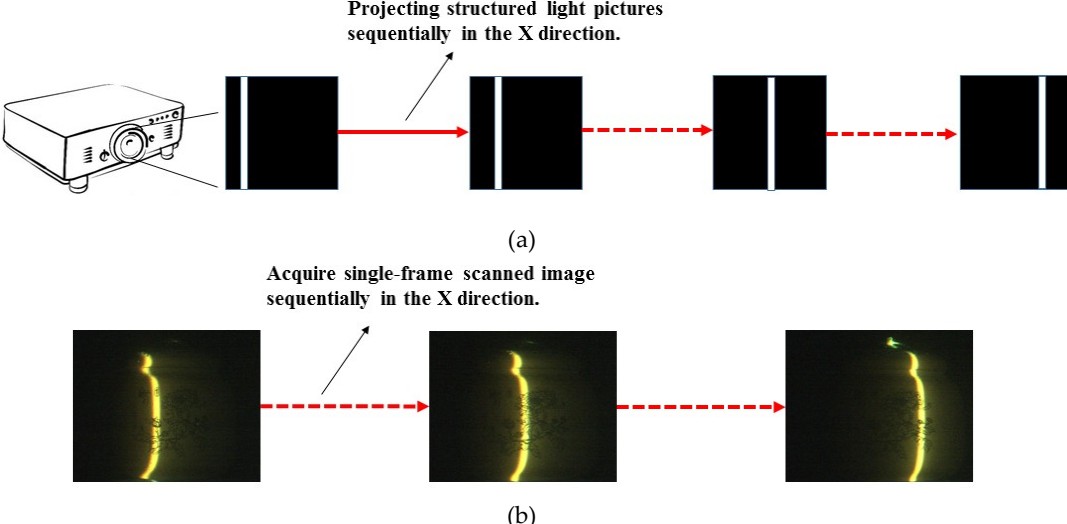

**Figure 3.** (**a**) Scheme of sequential illumination process. (**b**) Sequential frame images of the real object during scanning illumination.

## 3. Removal of Back Scattered Light

Eliminating back scattered light is a critical requirement in underwater imaging systems, for which a few techniques have been well developed [11–13]. The synchronous scanning system works with a collimated laser beam (or stripe) and an opto-electric sensor with a narrow field of view; both are synchronously scanned to limit back scattered light within the small intervening volume during image scanning. Alternatively, as described in [10], the back scattered light generated by stripe lighting can be computed, if the 3D reconstruction of the scene is obtained and the attenuation coefficient is evaluated. Thus, the scene appearance without scattered light can be evaluated for each lighted point of the object. Both approaches, however, require either complex system constructions or prior measurements of the optical properties of the medium.

As illustrated in Figure 1, for stripe lighting in a scattering medium, the back scattered background appears as a light sheet from which brightness falls off exponentially along the projection direction, ending at the intersection of the surface. If the reflected light by the surface could yield to a discontinuity of brightness at the end of the light sheet, we may separate the flat brightness of the light sheet from the surface reflected light by detection of such discontinuity. However, the brightness of the back scattered light plane is usually mixed with surface light, so detection of such a discontinuity is not reliable. A straightforward approach is to employ a window mask on the image to cut off the back scattered light sheet from the surface light, which is the same principle as the synchronous scanning system. However, the field of view is limited by a virtual "aperture", rather than the physics-based setup of the opto-electric sensor in the synchronous scanning system [14].

The size of the window mask is application dependent, which is needed to cover the reflected surface light and to remove back scattered light as much as possible. As the coverage of the projected

striping illumination region is known, we can preset a threshold to first determine the starting point of the window at the empty side of the light stripe (far end of the back scattered light sheet), then determine the ending point (near end of the back scattered light sheet) according to the coverage of the striping illumination region. Considering that noise and texture of the surface albedo may induce uncertainties in the reflected surface light, we apply the smoothing filter before masking the support of the surface light by thresholding. Figure 4 shows an example of the determination of a window mask for a single frame image. Figure 5 shows results for synthesized images before and after the removal of back scattered light.

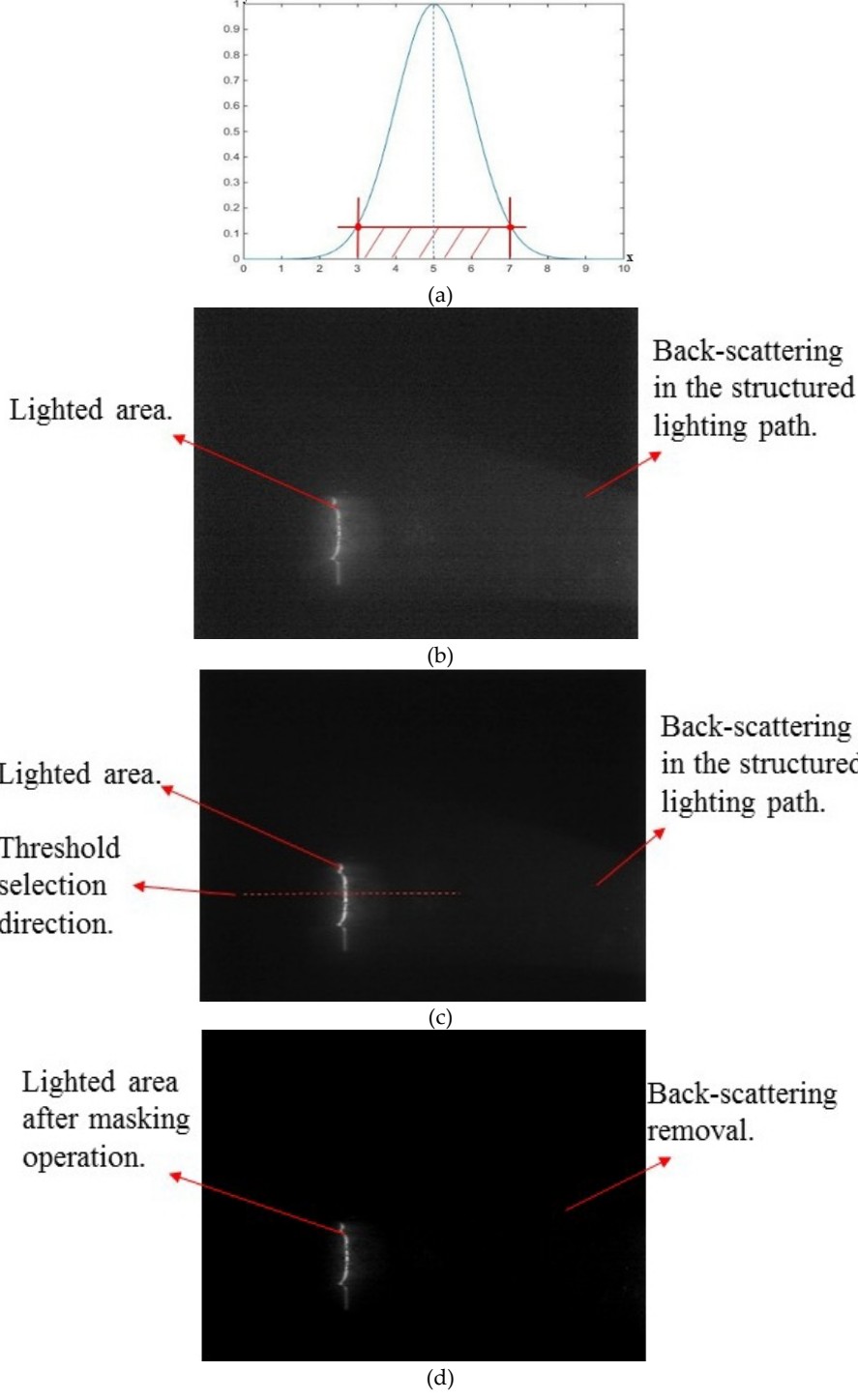

**Figure 4.** *Cont.*

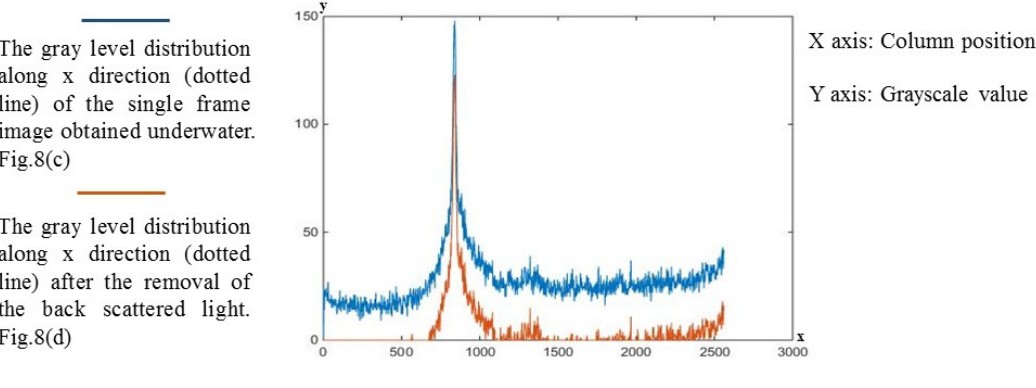

The gray level distribution along x direction (dotted line) of the single frame image obtained underwater. Fig.8(c)

The gray level distribution along x direction (dotted line) after the removal of the back scattered light. Fig.8(d)

X axis: Column position

Y axis: Grayscale value

(e)

**Figure 4.** Determination of window mask. (**a**) Transverse intensity distribution of the projecting stripe light and the preset threshold of the window. (**b**) Single frame image obtained underwater. (**c**) Single frame image after applying smoothing filter to (**b**). Along the dotted line, the starting and ending points of the mask are determined accordingly with respect to this line. (**d**) Frame image after the removal of back scattered light. (**e**) Gray level distribution along the *x* direction (dotted line) before and after the removal of backscattering.

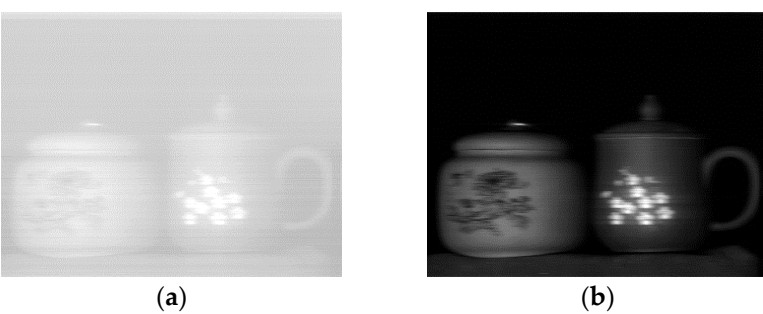

(**a**)　　　　　　　　　　　　　　　　(**b**)

**Figure 5.** Sample synthesized image. (**a**) Result before the removal of back scattered light. (**b**) Result after the removal of back scattered light.

Although the frame image with removal of back scattered light is not the computed exact scene image, as described previously in the method of [10], the narrow field of view by the imposed virtual aperture can significantly eliminate back scattered light in the case of striping illumination. It should be noted that the masking operation inevitably involves loss of the scene on each frame image. However, such a loss of the scene can be recognized as a result of the change of illumination function *I(x)* in the model of Equation (5). If we can properly estimate the changed illumination function through blind deconvolution, we can fully recover the albedo of the object.

## 4. Experimental Results

Following the descriptions in Sections 2 and 3, the steps of the system implementation are described below:

(i) Project the designed illumination pattern and record the frame image synchronously;

(ii) Complete the scanning process by sequentially translating the illumination pattern with 1-pixel displacement at the projector and obtain the sequential frame images accordingly;

(iii) Apply the virtual aperture to each frame image to remove back scattered light;

(iv) Complete the convolutional integration by integrating each frame image in one dimension, i.e., to sum up the row of pixel values to output a single pixel value, and one frame image outputs one column of the image *g(s)* of Equation (5); all frame images give rise to the complete *g(s)*;

(v) Apply the deconvolution routine to recover the albedo *O(x)* of Equation (5); the initial estimate of *I(x)* is given by the designed illumination pattern.

In our experiments, objects were placed in a $150 \times 150 \times 300$ cm$^3$ glass tank. The scattering medium was emulated using water mixed with different quantities of milk and aluminum hydroxide. The CCD video camera is a GO-5000-PGE camera with maximum resolution of 2560*2048 and a maximum frame rate of 22. The projector is an EPSON CH-TW6700W with 1.6x zoom lens and resolution of 1080 p ($1920 \times 1080$). The lamp brightness is 3000 lumens. A CCD video camera and a DLP projector were placed outside the tank. The experimental setup and examples of the frame image in water of varying turbidity are shown in Figure 6.

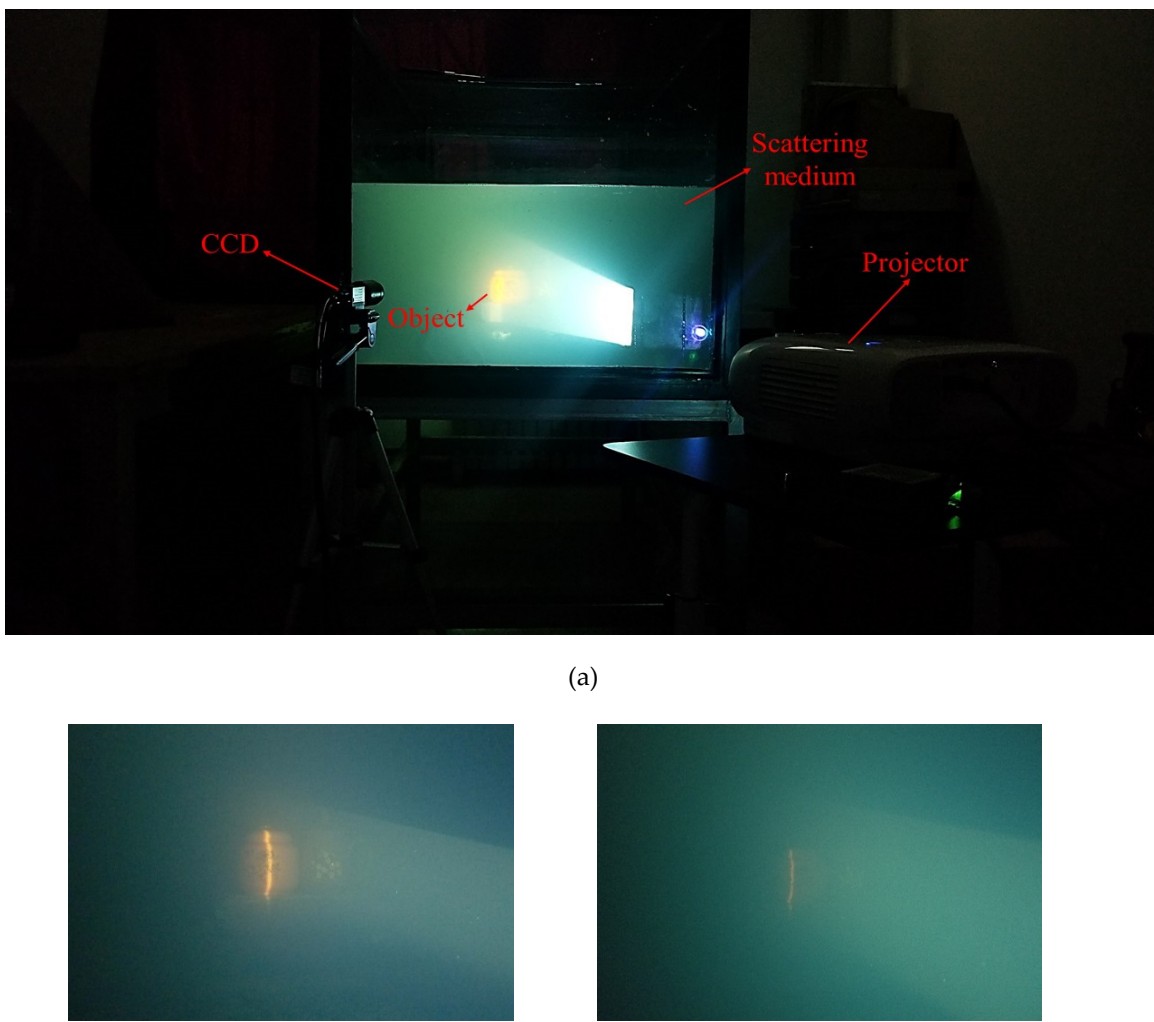

(a)

(**b**)            (**c**)

**Figure 6. (a**) Experimental setup; (**b**) Scanning image with light turbidity water; (**c**) Scanning image with heavy turbidity water.

The experimental results were obtained following the implementation steps as listed above. For image recovery, the well-developed blind deconvolution algorithm of Lucy–Richardson method (L–R) was applied [15–17].

The results of image recovery with light turbidity of water are shown in Figure 7. Considering the wavelength-dependent transmission in underwater imaging, we also applied the algorithm to RGB channels, and the colored versions of image recovery are given below the gray-scale versions. Figure 8 shows the results of image recovery with heavy turbidity of water.

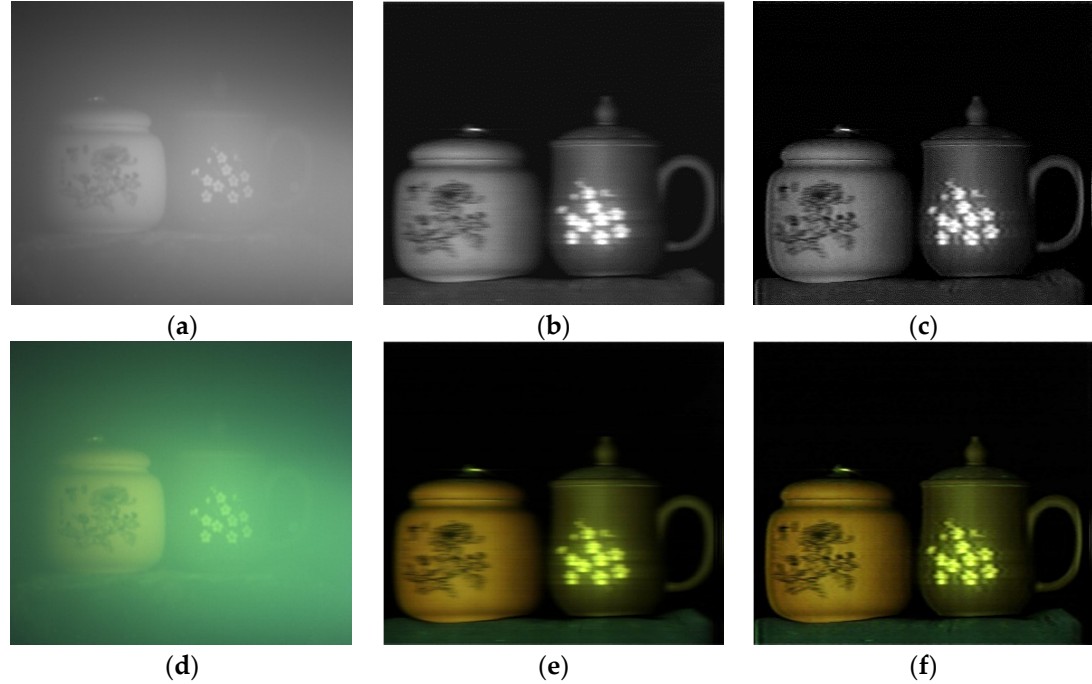

**Figure 7.** Experimental results of image recovery with light turbidity of water. (**a**) Floodlit gray level image; (**b**) Synthesized image after the removal of back scattered light; (**c**) Result of blind deconvolution using the Lucy–Richardson (L–R) method; (**d**–**f**) Colored versions associated with the same implementations as of (**a**–**c**) with respect to RGB channels, respectively.

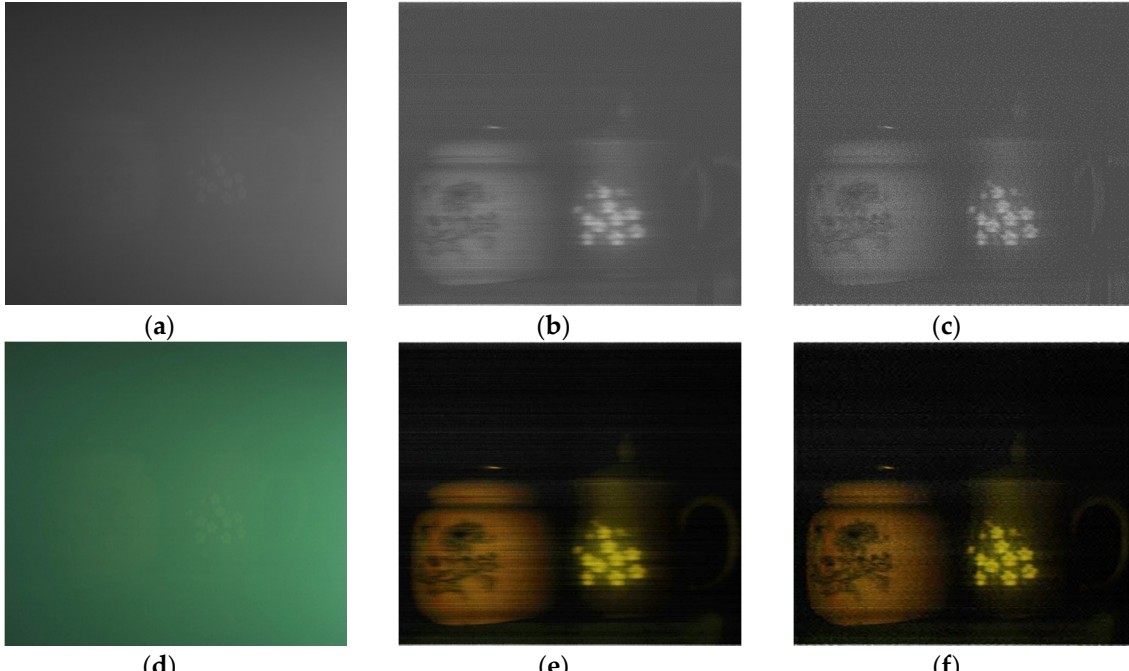

**Figure 8.** Experimental results of image recovery with heavy turbidity of water. (**a**) Floodlit gray level image; (**b**) Synthesized image after the removal of the back scattered light; (**c**) Result of blind deconvolution using the L–R method; (**d**–**f**) Colored versions associated with the same implementations as of (**a**–**c**) with respect to RGB channels, respectively.

Compared to the floodlit image, the recovered images using the proposed method have a significantly improved image contrast, even with heavy turbidity of the water. The improvement

is twofold: Removal of the back scattered light by using the virtual aperture to each frame image effectively suppresses the back scattered background in the synthesized image; the subsequent image recovery eliminates the blur of the albedo image by collecting the forward scattered light. Moreover, the colored version of the image can be recovered, pleasing to the visual inspection. It is noticed that, since light attenuation underwater is wavelength depended, the colored recovered image can also infer the spectral nature of the medium. The experimental results show that the recovered object albedo looks reddish compared to its appearance in the air, implying that the red channel has better penetration in turbid water than green and blue channels, at least in the emulated medium in our experiments.

## 5. Conclusions

This paper presents a new approach to underwater image recovery using sequential striping illumination, which can tackle both forward scattering and back scattering problems. The imaging process is modeled on the implementation of a convolution integration, and image acquisition is issued as image recovery by deconvolution. The method has the advantages of integrating forward scattered light into a recovered image so as to eliminate image blur. Analogous to the mechanism of the synchronous scanning system, back scattered light is eliminated by a virtual "aperture", which can be realized by window masking each frame image. Although removal of the back scattered light was inaccurate, it performed effectively in underwater experiments. Results of image recovery verified the effectiveness of the proposed method: Visibility of the recovered object image is significantly better than that of the floodlit image in turbid water medium, although a colored version of image recovery is also available.

In the proposed approach, image acquisition does not require 3D calibration of the system. The 3D reconstruction of the scene has not been considered in this paper. Obviously, 2D image recovery facilitates the modality of 3D reconstruction; image recovery enables the separation of the striping illumination pattern from the surface albedo, yielding to more reliable detection of the center of the projected stripe.

Finally, it should be noted that the width of the lighting stripe affects the image recovery by deconvolution. A broadly distributed illumination function $I(x)$ inevitably conducts the degradation of image deconvolution by using the model of Equation (5), which is a common problem mathematically. It is expected to specify a more sophisticated lighting pattern to improve the robustness of image recovery, which will be considered in future work.

**Author Contributions:** Conceptualization, G.W.; methodology, B.G. and G.W.; software, B.G.; validation, B.G; formal analysis, B.G. and G.W.; investigation, B.G.; resources, G.W.; data curation, B.G.; writing—original draft preparation, G.W.; writing—review and editing, B.G. and G.W.; visualization, B.G.; supervision, G.W.; project administration, G.W.; funding acquisition, G.W.

**Funding:** This research was funded by the National Natural Science Foundation of China, grant number 61571407.

**Acknowledgments:** This paper has undergone English language editing by MDPI. The text has been checked for correct use of grammar and common technical terms, and edited to a level suitable for reporting research in a scholarly journal.

**Conflicts of Interest:** We declare no conflict of interest.

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
