# Peer review of "Underwater 2D Image Acquisition Using Sequential Striping Illumination"

_applsci, doi:10.3390/app9112179_

Round 1

Reviewer 1 Report

General comments
==================

The manuscript describes the technique of imaging objects in a turbid medium scene using sequential imaging of an illuminated strip scanned across the objects. The study provides a mathematical treatment for the recovery of the underlying image (surface albedo) as a deconvolution operation on the captured blurred image. A heursistic method is presented to supress the effect of back scattered light from the turbid medium, which aids in the recovery of details of the deconvolved image. Examples of images taken in a turbid underwater setting are provided to motivate the techniques described. The removal of the back scattering is a nice treatment, but it is actually missing from the title of the manuscript itself. If possible, I would suggest to highlight it.

Overall, I find the study is scientifically sound. I commend the author's efforts towards providing both a mathematical description of the operation as well as several examples of the method on real images. I think the study is well motivated by the examples, but I would argue (below) that some more details are required to allow for scientific replication/clarity. I think most of the information I considered 'missing' is easily available to the authors -- it just needs to be integrated and structured into the text to benefit the reader.

The manuscript text is generally well written, but it does suffer from awkward phrases or formulations in english. I would urge an english proofreading effort before resubmitting.

The minor linguistic problems combined with the lack of details in many descriptions or figure captions, makes the content of the study rather difficult to understand in one reading. I provide some suggestions below to improve readability:

- The structure of the manuscript leaves the description of the imaging system till the very end. So I struggled to see how the theoretical text mapped to practical operations till I saw the imaging setup, and had to go through the text again. I suggest the manuscript could be organized as Intro-Theory-Materials-Methods-Results-Conclusion. The Materials section describes the setup, the Methods section should describe how the images are actually acquired and then processed.

 - Apart from the introduction, a large part of the text has no references. More focussed & appropriate reference citations would help motivate the study better.

- The figure captions are very sparse and provide little aid in mapping the contents of the figure to the text of the manuscript. I suggest fuller and descriptive figure captions.

- The actual scanning procedure is not described in detail, so several details are not clear about the practicalities of how a 2D image is reconstructed from a series of 2D CCD sensor images. Does the imager move with the scanning stripe? Is the light stripe source rotated, or linearly translated with respect to the scene? A Materials & Method's section should describe the details of this. The actual instrumentation details of the CCD camera or light projector are also not provided.

- Figure 1 is a great place to explain the details of the technique, but falls short of it in its current form. I would expect this figure to be annotated with "x" and "t" and other elements, such as back scattered light, reflected light, etc, describing the optics of the situation.

- This may be a minor point, but the use of "t" as a variable in the equations to denote a spatial position threw me off. General convention is to use to for a temporal dimension, and in this case there is a temporal aspect to the acqusition (scanning). So it took me a couple of readings to clarify t is a position. I would suggest to rename t to something else, and also show these variable on Figure 1.

- The actual workflow is also rather opaque. The mathematical treatment is good, but an algorithmic pseudo-code style description of the data treatment should be presented, esp because the workflow has several heuristic steps such as the virtual aperture or "window masking", which are missing descriptions.

- I suggest to merge Figure 6 & 7 and Figure 8&9 into single figures. Their captions are almost identical.

- Some minor a priori knowledge is required for this method to work, such as the size of the projected stripe on the imaged object plane. It would be good to list out these fairly simple pieces of information one must first collect clearly in the Methods/Theory description, as currently it pops up at various points in the text.

- What is the 'depth of field' of this techqnique? If one takes a very obliquely aligned object to the projected plane (going from near-field to far field), how well does this technique work? Where does it break down?

Specific comments
==================

L28: Six references (1-6) are simply provided in a line. It is better to provide a better motivation through explanation and include the references where appropariate. 2-3 references at a time.

L38: Two modalities "sequential" and "single shot" are mentioned, but not really explained what they are before describing their advantages/disadvantages. This should be reformulated, along with lines L46-50.

L43: Reference missing for "...with complex patterns are indistinguishable due to the poor image qualities".

L64-L74: While the content is good, it could be structured better. It reads partly like mathods, partly like conclusions and then has a trailing sequence of references at the very end. I suggest to move appropriate parts of it to other sections.

L83-132: This whole section would greatly benefit from clear annotation of Figure 1 with the elements of the equations being presented. A fuller figure caption should also help. At this point, I generally missed the overall step of how one goes from a "single" acqusition by the imager, to the 'stitching' together into a 2D scene through the "scanning action". This should be described so that a reader can understand how the theory maps to practice.

L106: What does "integrating the sequential stripe-lighted images" actually mean in practice?

Section 3: A pseudo-code or algorithmic description  of the treatment will help. Also missing are descriptions and citations to explain what "window masking" means, and what window does it refer to? What sort of smoothing operation was applied with what parameters? How is the transverse "support of the projecting light known" in the first place -- other a priori meausurements are required?

Figure 2: Panel a is missing labels for the x & y axes. The images in panels b, c and d provide a qualiative assessment of the backscatter removal operation -- but an understanding of its effect on the actual signal would help more. I recommend to add another panel where the pixel levels (along the dotted line) from panel b) c) and d) are shown together to demonstrate the 'removal of back scatter'. Later text states that it is not fully accurate, but good enough -- this would be a good way to show a quantitative measure of the treatment.

L174-176: I did not understand this. Could you please explain it better?

L181: How was the scanning performed? Details missing.

L187: What does "concerned with 3D measurement" mean? I suppose some comments about the assumptions/requirements about the 3D structure of the objects supported by this imaging technique should be mentioned. If an object is highly curved or edged, wouldn't the projected light sheet be no longer symmetric (required for deconvolution assumption) on the object?

Fig8&9: The RGB results are impressive. It would be good to include a description of the spectral nature of the light source. It seems that the turbid medium does not seem to have any spectral nature to its attenuation?

Author Response

Response to Reviewer 1 Comments

Point 1:  The manuscript describes the technique of imaging objects in a turbid medium scene using sequential imaging of an illuminated strip scanned across the objects. The study provides a mathematical treatment for the recovery of the underlying image (surface albedo) as a deconvolution operation on the captured blurred image. A heursistic method is presented to supress the effect of back scattered light from the turbid medium, which aids in the recovery of details of the deconvolved image. Examples of images taken in a turbid underwater setting are provided to motivate the techniques described. The removal of the back scattering is a nice treatment, but it is actually missing from the title of the manuscript itself. If possible, I would suggest to highlight it.

Response 1: We try to change the title of the manuscript to ‘Underwater Back-scattering Removal and Image Recovery Using Sequential Striping Illumination’, including the description of the method of removing back scattering.

Point 2: The structure of the manuscript leaves the description of the imaging system till the very end. So I struggled to see how the theoretical text mapped to practical operations till I saw the imaging setup, and had to go through the text again. I suggest the manuscript could be organized as Intro-Theory-Materials-Methods-Results-Conclusion. The Materials section describes the setup, the Methods section should describe how the images are actually acquired and then processed.

Response 2: We reorganized the manuscript structure as Intro-Theory-Materials-Methods-Results-Conclusion according to expert suggestion.

Point 3: Apart from the introduction, a large part of the text has no references. More focussed & appropriate reference citations would help motivate the study better.

Response 3: We added some new references.

Point 4: The figure captions are very sparse and provide little aid in mapping the contents of the figure to the text of the manuscript. I suggest fuller and descriptive figure captions.

Response 4: We revised the figure captions.

Point 5: The actual scanning procedure is not described in detail, so several details are not clear about the practicalities of how a 2D image is reconstructed from a series of 2D CCD sensor images. Does the imager move with the scanning stripe? Is the light stripe source rotated, or linearly translated with respect to the scene? A Materials & Method's section should describe the details of this. The actual instrumentation details of the CCD camera or light projector are also not provided.

Response 5: We added experimental setup scheme in the Materials section. The experimental implementation process is described in detail.

Point 6: Figure 1 is a great place to explain the details of the technique, but fallsshort of it in its current form. I would expect this figure to be annotated with"x" and "t" and other elements, such as back scattered light, reflected light, etc, describing the optics of the situation.

Response 6: We modified the description in Figure 1 in more details.

Point 7: This may be a minor point, but the use of "t" as a variable in the equations to denote a spatial position threw me off. General convention is to use to for a temporal dimension, and in this case there is a temporal aspect to the acquisition (scanning). So it took me a couple of readings to clarify t is a position. I would suggest to rename t to something else, and also show these variable on Figure 1.

Response 7: We eenamed t to s, and add this variable to Figure 1.

Point 8: The actual workflow is also rather opaque. The mathematical treatment is good, but an algorithmic pseudo-code style description of the data treatment should be presented, esp because the workflow has several heuristic steps such as the virtual aperture or "window masking", which are missing descriptions.

Response 8: We added workflow in the Results section.

Point 9: I suggest to merge Figure 6 & 7 and Figure 8&9 into single figures. Their captions are almost identical.

Response 9: Figure 6 & 7 and Figure 8&9 respectively represent the experimental results in different turbidity water bodies. The turbidity of water affects the light propagation. The turbidity discussion also verifies the adaptability of the method in different underwater environments.

We merged Figure 6 & 7 and Figure 8&9 into single figures in the revised draft.

Point 10: Some minor a priori knowledge is required for this method to work, such as the size of the projected stripe on the imaged object plane. It would be good to list out these fairly simple pieces of information one must first collect clearly in the Methods/Theory description, as currently it pops up at various points in the text.

Response 10: We listed some prior information in the Materials section.

Point 11: What is the 'depth of field' of this technique? If one takes a very obliquely aligned object to the projected plane (going from near-field to far field), how well does this technique work? Where does it break down?

Response 11: At present, we have only verified the method in the experimental water tank. The actual application depth depends on the intensity of the scanning light source. Because the energy of the structured light source is more concentrated, the propagation range is better than the traditional flooding method.

The position of the CCD and the projector is fixed during the imaging process. The scanning process is realized by projecting the PPT image moving sequentially along the X direction frame by frame. The overall imaging system is fixed. The experiment is reproducible, and the replacement target can theoretically achieve the same experimental results.

The absorption and scattering of water causes the energy of the light wave to decay. When the distance between the light source and the target exceeds the energy of the light source, the system will no longer be able to obtain the target information.

Specific comments:

Point 12: L28: Six references (1-6) are simply provided in a line. It is better to provide a better motivation through explanation and include the references where appropriate. 2-3 references at a time.

Response 12: We reorganized statements and references.

Point 13: L38: Two modalities "sequential" and "single shot" are mentioned, but not really explained what they are before describing their advantages/disadvantages. This should be reformulated, along with lines L46-50.

Response 13: We reformulated statements with explanation.

Point 14: L43: Reference missing for "...with complex patterns are indistinguishable due to the poor image qualities".

Response 14: We revised the statements and added appropriate references.

Point 15: L64-L74: While the content is good, it could be structured better. It reads partly like methods, partly like conclusions and then has a trailing sequence of references at the very end. I suggest to move appropriate parts of it to other sections.

Response 15: We revised the statements and moved parts of the description in section 2.

Point 16: L83-132: This whole section would greatly benefit from clear annotation of Figure 1 with the elements of the equations being presented. A fuller figure caption should also help. At this point, I generally missed the overall step of how one goes from a "single" acquisition by the imager, to the 'stitching' together into a 2D scene through the "scanning action". This should be described so that a reader can understand how the theory maps to practice.

Response 16: We modified the description in Figure 1.

Point 17: L106: What does "integrating the sequential stripe-lighted images" actually mean in practice?

Response 17: The scanning process is realized by sequentially projecting the programmed PPT image of the designed illumination pattern, each of which has 1-pixel displacement of the center of the illumination pattern; frame images are synchronously recorded by the CCD, respectively. For each frame image, the integration is taken in one dimension, i.e., to sum up the row pixel values to output a single pixel value; one frame image outputs one column of the synthesized image. Consequently, all frame image give rise to the whole synthesized image column by column.

Point 18: Section 3: A pseudo-code or algorithmic description of the treatment will help. Also missing are descriptions and citations to explain what "window masking" means, and what window does it refer to? What sort of smoothing operation was applied with what parameters? How is the transverse "support of the projecting light known" in the first place -- other a priori measurements are required?

Response 18: We added the algorithmic description in section 3. The window masking refers to a 1-D rectangular masking operation, which location is determined row by row accordingly, as described in section 3.

Point 19: Figure 2: Panel a is missing labels for the x & y axes. The images in panels b, c and d provide a qualitative assessment of the backscatter removal operation -- but an understanding of its effect on the actual signal would help more. I recommend to add another panel where the pixel levels (along the dotted line) from panel b) c) and d) are shown together to demonstrate the 'removal of back scatter'. Later text states that it is not fully accurate, but good enough -- this would be a good way to show a quantitative measure of the treatment.

Response 19: We Changed Figure 2. We added pixel level distribution along the dotted line to show a Quantitative comparison of the results of before and after backscatter removal

Point 20: L174-176: I did not understand this. Could you please explain it better?

Response 20: The loss of the scene is due to the masking operation, which can be recognized as the result of the change of the illumination function in the convolution model. If we can properly estimate the changed illumination function through blind deconvolution, we thus can fully recover the albedo of the object.

Point 21: L181: How was the scanning performed? Details missing.

Response 21: We added workflow and experimental set scheme in the Materials section.

Point 22: L187: What does "concerned with 3D measurement" mean? I suppose some comments about the assumptions/requirements about the 3D structure of the objects supported by this imaging technique should be mentioned. If an object is highly curved or edged, wouldn't the projected light sheet be no longer symmetric (required for deconvolution assumption) on the object?

Response 22: The structured light system used in this paper is capable of acquiring three-dimensional information of underwater targets. Therefore, the original manuscript wants to highlight that. But this paper only deals with the image acquisition problem. If an object is highly curved or edged, there indeed would be deformation of the light stripe, causing the local deformation of the recovered object.

Point 23: Fig8&9: The RGB results are impressive. It would be good to include a description of the spectral nature of the light source. It seems that the turbid medium does not seem to have any spectral nature to its attenuation?

Response 23: We added an analysis of the influence water turbidity on the spectral nature. Briefly, the Red channel has less attenuation than Green and Blue channels in highly turbid water.

Reviewer 2 Report

The authors tried to propose an approach for 2D image recovery using structured light in a scanning mode underwater. They used the integration of every sequential stripe-lighted images, to generate a synthesized image to be modeled as the convolution of the illumination function and the surface albedo.

In my opinion, this manuscript has these very good points:

- The research subject is intriguing and might absorb many readers in several fields;

- If the explanations increase adequately it would be very interesting;

Also, there are some suggestions that would increase the strength of the paper which is listed bellows;

-          The major issue of this approach is related to lack of analyses and results. Even the conclusion of this approach is obscure to readers.

-          The authors need to really put their affords to fix this manuscript. Despite the interesting topic the authors partially failed to present it properly.

-          The first thing that authors must fixed is to describe much clearer what is 2D imaging acquisition to readers, since this might be specific topic and not very popular, to everyone imaging is 2D.

-          Please provide workflow before start your descriptions.

-          The authors need to increase their imaging samples. What I am seeing is one set of acquisition from the beginning to end (this is enough for me to reject the manuscript!)

-          Experimental set scheme is really needed, I have no idea what is going on from Figure4!

-          P5,L163, Authors stated “(a) The sectional intensity distribution of the projecting 163 stripe light and the preset threshold of the window.” Explicit explanation is required!

-          Eq.1: f (x,t) = I (x − t)O(x), L87 authors stated: “noted I(x), where x is along the scanning direction.” Point scanner? Line scanner?

-          I recommend to use these articles as reference and model of using math notations:

o    Visual Servoing for Underwater Docking of an Autonomous Underwater Vehicle with One Camera, Oceans 2003. Celebrating the Past ... Teaming Toward the Future (IEEE Cat. No.03CH37492)

o    Hierarchical segmentation of urban satellite imagery, International Journal of Applied Earth Observation and Geoinformation 2014;

o    Quantitative Photogrammetric Analysis of Digital Underwater Video Imagery, IEEE JOURNAL OF OCEANIC ENGINEERING, VOL. 22, NO. 2, APRIL 1997

o    Underwater Optical Imaging: Status and Prospect, Oceanography, VoL 14, No. 3/2001.

o    A novel fuzzy based method for 3D buildings modelling in urban satellite imagery, 2011 IEEE Conference on Open Systems

- It is not clear that why the authors mentioned the Figure 8, please elaborate more about this figure.

Some editorial points:

- Authors must concretely modify their figures, current condition is not even adequate!

- Please send your manuscript to an official proofreading to increase the readability of your manuscript;

In overall, this work is nice to be published after intense modifications, Therefore, I ask for re-submitting the manuscript to give a chance to the authors to modify their manuscript.

Thank you

Author Response

Point 1: The major issue of this approach is related to lack of analyses and results. Even the conclusion of this approach is obscure to readers.

Response 1: We revised the statements of the manuscript. We Added analysis in the experimental settings and results section, and further described the experimental conclusions.

Point 2: The authors need to really put their affords to fix this manuscript. Despite the interesting topic the authors partially failed to present it properly.

Response 2: We revised the whole manuscript and reorganized the manuscript structure as Intro-Theory-Materials-Methods-Results-Conclusion.

We also Added descriptions to the experimental implementation process and analyzed the results in more detail. We Added references to help to understand some contexts.

Point 3: The first thing that authors must fixed is to describe much clearer what is 2D imaging acquisition to readers, since this might be specific topic and not very popular, to everyone imaging is 2D.

Response 3: The structured light system used in this paper is capable of acquiring three-dimensional information of underwater targets. Therefore, the original manuscript wants to highlight that this paper only deals with the image acquisition problem. According to the reviewer’s comments, we think that the term “2D” might induce confusions to the readers, so we canceled the term “2D” in the title.

Point 4: Please provide workflow before start your descriptions.

Response 4: We added workflow in the Results section.

Point 5: The authors need to increase their imaging samples. What I am seeing is one set of acquisition from the beginning to end (this is enough for me to reject the manuscript!)

Response 5: The experimental process in this paper is mainly a theoretical verification experiment, not an applied experiment, which mainly verifies the feasibility of the proposed algorithm theory.

The experimental results verify the validity of the proposed method. The position of the CCD and the projector is fixed during the imaging process. The scanning process is realized by projecting the PPT moving sequentially along the X direction frame by frame. The overall imaging system is stable. The experiment is reproducible, and the replacement target can theoretically achieve the same experimental results, so we just used one set in the paper.

Point 6: Experimental set scheme is really needed; I have no idea what is going on from Figure4!

Response 6: We added experimental set scheme in the Materials section. The experimental implementation process is described in detail.

Point 7: P5,L163, Authors stated “(a) The sectional intensity distribution of the projecting 163 stripe light and the preset threshold of the window.” Explicit explanation is required!

Response 7: We revised that statement. We added a description of the method to process removal of the backscattering.

Point 8: Eq.1: f (x,t) = I (x − t)O(x), L87 authors stated: “noted I(x), where x is along the scanning direction.” Point scanner? Line scanner?

Response 8: The x direction is marked in Figure 1. It refers to line scanner (striping illumination).

Point 9:I recommend to use these articles as reference and model of using
math notations:
o Visual Serving for Underwater Docking of an Autonomous Underwater Vehicle with One Camera, Oceans 2003. Celebrating the Past ... Teaming Toward the Future
(IEEE Cat. No.03CH37492)
o Hierarchical segmentation of urban satellite imagery, International Journal of Applied Earth Observation and Geoformation 2014;
o Quantitative Photogrammetric Analysis of Digital Underwater Video Imagery, IEEE JOURNAL OF OCEANIC ENGINEERING, VOL. 22, NO. 2, APRIL 1997
o Underwater Optical Imaging: Status and Prospect, Oceanography, VoL 14, No. 3/2001.
o A novel fuzzy based method for 3D buildings modelling in urban satellite imagery, 2011 IEEE Conference on Open Systems

Response 9: We Found and read the literature recommended by experts, and used these articles as reference in the revised manuscript.

Point 10: It is not clear that why the authors mentioned the Figure 8, please elaborate
more about this figure.

Response 10: We show the colored version since we thought that RGB images are more pleasing to the subjective vision of humans. Moreover, the light attenuation in underwater is wavelength depended, particularly in turbid water. Although the projecting light is white, image recovery by the proposed method can be implemented   in each RGB channel, each of which has in fact different scattering/attenuation properties. The colored version can also infer the spectral nature of the medium.

Point 11:Authors must concretely modify their figures, current condition is not even
adequate!

Response 11: We revised the figure captions.

Point 12: Please send your manuscript to an official proofreading to increase the readability of your manuscript;

Response 12: We asked the official proofreading(MDPI) to increase the readability of our revised manuscript.

Round 2

Reviewer 2 Report

Underwater 2D Image Acquisition Using Sequential Striping Illumination

Great afford by authors.The authors nicely tried to reply my comments well and I am convinced to accept the manuscript but there are some very small editorial points the should be considered before.

Please follow these instructions to make your figures better:

·         Figure 1 and figure 2 should be in the size of text. Make it big to the size of your text:

o   if you are using LaTex use following command “\includegraphics[width= 1\linewidth]{ Fig1.png}”

o   If you are using word, just make the figure bigger to the size of your text

And it must be at the top of the page.

·         Merge figure 3 and Figure 4 in one figure. Then in the images make with arrow and text what do you mean by these lines.

·         In Figure 5, please write in the images make with arrow and text what do you mean by showing these images. The text of Figure 5.e is not readable, please make the bigger. Increase the quality of your images, current resolution is very low for printing.

·         Please merge Figure 6, Figure 9, and figure 10 and make one figure with a good caption (you already did some good job in that but needs to be better).

·         Please merge Figure 7 and Figure 8 also, Make Figure 7 bigger to the size of text. Then you need to increase the font text indicating the explanations regarding the experiment.

Formulations:

·         In P3, 113-115, you mentioned: “In underwater imaging, however, the forward scattering occurs during light transmission from the object to the camera, whose imaging effect can be expressed by the point spread function (PSF), h(x), whereas back scattering induces added background noise, n(x). Thus, the general image formation is:”

In you formulation Equation (2), you did not add n(x)!!! Why? Something is missing.

·         Please carefully check your formulations, there should be nothing missing.

·         Use lower case in your Keywords.

In overall, I ask for minor revision to give a chance to the authors to finally modify their manuscript before I accept the manuscript.

Thank you

Author Response

Response to Reviewer 2 Comments

Point 1: Figure 1 and figure 2 should be in the size of text. Make it big to the size of your text: If you are using word, just make the figure bigger to the size of your text. And it must be at the top of the page.

Response 1: We revised the figures’ size.

Point 2: Merge figure 3 and Figure 4 in one figure. Then in the images make with arrow and text what do you mean by these lines.

Response 2: We merged figure 3 and Figure 4 in one figure, added text in the images to explain the meaning of the lines.

Point 3: In Figure 5, please write in the images make with arrow and text what do you mean by showing these images. The text of Figure 5.e is not readable, please make the bigger. Increase the quality of your images, current resolution is very low for printing.

Response 3: We added text in the images to explain the meaning of the images.

          Make Figure 5.e bigger.

          Increase the quality of our images    

Point 4: Please merge Figure 6, Figure 9, and figure 10 and make one figure with a good caption (you already did some good job in that but needs to be better).

Response 4: Figure 6 demonstrated the effectiveness of backscattering removal method proposed by this paper.

         Figure 9, and figure 10 showed the results of different turbidities.

            We think that the current layout of these three figures is reasonable and requests to maintain the current situation.

Point5: Please merge Figure 7 and Figure 8 also, Make Figure 7 bigger to the size of text. Then you need to increase the font text indicating the explanations regarding the experiment.

Response 5: We merged figure 7 and Figure 8 in one figure; Make Figure 7 bigger.

In the second section of the paper, we discuss the implementation of the experiment and the system construction. In the experimental part, we discuss the experimental application equipment and give the experimental scene. We think that the current description can clearly describe how the method is implemented.

Formulations:
Point6:In P3, 113-115, you mentioned: “In underwater imaging, however, the forward scattering occurs during light transmission from the object to the camera, whose imaging effect can be expressed by the point spread function(PSF), h(x), whereas back scattering induces added background noise, n(x).Thus, the general image formation is:”
In you formulation Equation (2), you did not add n(x)!!! Why? Something is missing.

Response 6: In the previous article we mistakenly used the letter n in the paper, we fixed this error.

Point7: Please carefully check your formulations, there should be nothing missing.

Response 7: We checked our formulations, there was nothing missing.

Point8: Use lower case in your Keywords.

Response 8: We used lower case in our Keywords. 
